# 30-Year Development of Inactivated Virus Vaccine in China

**DOI:** 10.3390/pharmaceutics15122721

**Published:** 2023-12-02

**Authors:** Jinrong Shi, Ailin Shen, Yao Cheng, Chi Zhang, Xiaoming Yang

**Affiliations:** 1National Engineering Technology Research Center for Combined Vaccines, Wuhan 430207, China; jrshi2002@sohu.com (J.S.); m17764008160@163.com (A.S.); cyao0622@gmail.com (Y.C.);; 2Wuhan Institute of Biological Products Co., Ltd., Wuhan 430207, China; 3China National Biotech Group Company Limited, Beijing 100029, China

**Keywords:** China, virus, inactivated vaccine, development

## Abstract

Inactivated vaccines are vaccines made from inactivated pathogens, typically achieved by using chemical or physical methods to destroy the virus’s ability to replicate. This type of vaccine can induce the immune system to produce an immune response against specific pathogens, thus protecting the body from infection. In China, the manufacturing of inactivated vaccines has a long history and holds significant importance among all the vaccines available in the country. This type of vaccine is widely used in the prevention and control of infectious diseases. China is dedicated to conducting research on new inactivated vaccines, actively promoting the large-scale production of inactivated vaccines, and continuously improving production technology and quality management. These efforts enable China to meet the domestic demand for inactivated vaccines and gain a certain competitive advantage in the international market. In the future, China will continue to devote itself to the research and production of inactivated vaccines, further enhancing the population’s health levels and contributing to social development. This study presents a comprehensive overview of the 30-year evolution of inactivated virus vaccines in China, serving as a reference for the development and production of such vaccines.

## 1. Overview of Inactivated Vaccines in China

In December 2019, the COVID-19 pandemic emerged, exerting a global impact [1]. The effective control of COVID-19 spread has been achieved through widespread administration of inactivated vaccines both domestically and internationally. This experience serves as a valuable reference for global vaccination efforts. The inactivated virus vaccine, employing a mature manufacturing technology, consists of viruses that have lost their activity. It stimulates the immune system to generate protective immunity against the virus. This vaccine approach has proven successful in safeguarding against diseases worldwide. The present paper reviews and synthesizes the 30-year development trajectory of inactivated vaccines in China, offering insights into the advancement and production of such vaccines.

Currently, China has approved several key inactivated vaccine products, including the Inactivated Tick-borne Encephalitis Vaccine, Inactivated Hemorrhagic Fever with Renal Syndrome Vaccine, Hepatitis A Vaccine, Inactivated Rabies Vaccine for Human Use, Inactivated Japanese Encephalitis Vaccine, Inactivated Poliomyelitis Vaccine, Inactivated Enterovirus Type 71 Vaccine, Quadrivalent Influenza Split Vaccine, and Inactivated 2019-nCoV vaccine. Over the past three decades, the cell substrates used for inactivated vaccine production in China have evolved from primary cells to more advanced types such as passage cells and human diploid cells. Advanced technologies such as bioreactor microcarrier suspension culture, chip carrier culture, bioreactor suspension culture, and shaker technology have replaced traditional methods such as single-layer cell culture and roller bottle culture. These advancements have enhanced the scale manufacturing capacity and quality of viral vaccines. The purification process of vaccines has also progressed, transitioning from simple microfiltration clarification to zone centrifugation and ultimately to the current chromatography purification. This evolution aligns with market demands for vaccine purity, sterility, and safety.

## 2. Inactivated Tick-Borne Encephalitis Vaccine

Tick-borne encephalitis is a rapid-onset viral infectious disease caused by the tick-borne encephalitis virus, primarily affecting the central nervous system [2]. In 1952, the virus strain responsible for tick-borne encephalitis was isolated from the brain tissue of an afflicted patient in Sanchazi, Linjiang town, Jilin Province, by the health epidemiology group of China Medical University. By 1953, the Changchun Institute of Biological Products Co., Ltd., China (referred to as Changchun Institute), had developed the tick-borne encephalitis vaccine using this strain, producing vaccine variants using mouse brain and chicken embryo tissue. In 1958, China initiated the development of an inactivated vaccine using chicken embryo cells. However, this vaccine exhibited inferior stability and immune effectiveness compared to the mouse brain-inactivated vaccine. In 1967, China transitioned to using hamster kidney cells for preparing the inactivated tick-borne encephalitis vaccine. The method involved inoculating the “Senzhang” strain onto a monolayer of hamster kidney cells, harvesting the viral solution after a specified incubation period, and subsequently inactivating the harvested viral solution with formaldehyde. While this vaccine demonstrated superior immune effectiveness compared to the chicken embryo cell inactivated vaccine, it had drawbacks such as low immunogenicity and a propensity for side effects.

In the 1990s, China initiated an independent development effort for the purified inactivated tick-borne encephalitis vaccine using hamster kidney cells. Biotechnological advancements, including continuous flow centrifugation, ultrafiltration concentration, and column chromatography purification, were employed in the preparation process. Various factors, such as incubation temperature and manufacturing technology, were optimized to extend the infection period of the “Senzhang” strain on hamster kidney cells, enabling continuous harvesting of the viral solution. This optimization facilitated the production of the purified inactivated tick-borne encephalitis vaccine. Through persistent exploration and enhancement, the Changchun Institute of Biological Products Co., Ltd. in China successfully developed the purified inactivated tick-borne encephalitis vaccine based on hamster kidney cells in 2001. Animal testing demonstrated the vaccine’s commendable immunogenicity. Clinical trials in 2003 further confirmed the safety and efficacy of the tick-borne encephalitis purified vaccine (hamster kidney cell), noting minimal side effects and adverse reactions. The vaccine exhibited a significant increase in antigen content, effectively safeguarding the human immune system and justifying mass-scale promotion and use. In 2004, the purified inactivated tick-borne encephalitis vaccine (hamster kidney cell) named “Sentaibao” received national certification as a new drug, along with a manufacturing number. It gained marketing approval in 2005, with a recommended basic immunization schedule of two doses at a 14-day interval. The antibody-positive conversion rate after immunization exceeded 85% [3]. This purified inactivated tick-borne encephalitis vaccine, originating in China, has been widely manufactured and utilized for many years, maintaining a stable immune effect. Its significant contribution to the prevention and treatment of tick-borne encephalitis in China is noteworthy. Table 1 summarizes the development of inactivated tick-borne encephalitis vaccine.

## 3. Inactivated Hemorrhagic Fever with Renal Syndrome Vaccine

Hemorrhagic fever with renal syndrome (HFRS), previously recognized as epidemic hemorrhagic fever, is an acute natural infectious disease caused by Hantavirus, primarily transmitted by rodents, particularly mice [4]. In China, HFRS is predominantly induced by Type I Hantanvirus (HTNV) and Type II Seoulvirus (SEOV). A total of five HFRS strains have been identified: Type I Z10 strain (1982), Type I LR1 strain (1983), Type II L99 strain (1983), Type II Z37 strain (1989), and Type I PS-6 strain (1994). Due to distinct clinical manifestations, epidemiological characteristics, main animal hosts, and, notably, antigenicity differences, the two types of hemorrhagic fever fail to offer mutual protection effectively. The preventive efficacy of monovalent vaccination falls significantly short of expectations. Consequently, a bivalent HFRS vaccine was developed in China based on the original monovalent vaccine, employing primary hamster kidney cells.

In 2002, Changchun Institute of Biological Products Co., Ltd. introduced the bivalent hemorrhagic fever with renal syndrome inactivated vaccine (hamster kidney cell). The hamster kidney cell vaccine utilized the SEO Hantavirus L99 isolated from the lung of the Jiangxi yellow hair mouse. The cells were acclimatized and passaged in hamster kidney cells, with a titer log TCID50/mL of 8.5. The primary hamster kidney cells were inoculated with the virus-containing cell medium, and after reaching peak virus proliferation, the medium was harvested, and cells were infected. Following freezing treatment and filtration, the cells were inactivated using 1/4000 formalin and augmented with 0.5 mg/mL Al(OH)_3_ as an adjuvant. Animal tests, including hamsters and rabbits, demonstrated significant protection against SEO (Type II) virus infection. Hamster protection tests also indicated significant defense against HTN (Type I) virus attack. Immunized animals generated neutralizing antibodies against the homologous Hantavirus, with some cross-reaction to HTN (Type I). Moreover, the vaccine induced significant cytotoxic T lymphocyte (CTL) activity, gamma interferon, and leukocyte interleukin-2 production in mice. Human lymphocyte proliferation response also improved after vaccination. After completing three doses on days 0, 15, and 1 year, 90% of vaccinated individuals retained positive neutralizing antibodies 33 months post-vaccination [5,6]. To address low production yield and contamination risks associated with hamster kidney cell vaccines, Royal (Wuxi) Biopharmaceuticals Co., Ltd. (China) substituted primary hamster kidney cells with Vero cells. The bivalent inactivated vaccine for hemorrhagic fever with renal syndrome (Vero cells) from the company was introduced to the market in 2003.

In 2005, Zhejiang Tianyuan Biopharmaceuticals Co., Ltd. (China) introduced a bivalent inactivated vaccine for hemorrhagic fever with renal syndrome (gerbil kidney cell). The gerbil kidney cell HTN (Type I) vaccine was formulated as follows: Primary gerbil kidney cells were inoculated with the Z10 strain (HTN Hantavirus), which were isolated from patients in Zhejiang Province. Simultaneously, primary gerbil kidney cells were inoculated with a mouse brain-acclimatized strain suspension. When the virus reached its peak hemagglutinin production during cultivation, the infected cells underwent ultrasound treatment, and the resultant mixture with the supernatant was centrifuged and inactivated using 1/4000 β-propanolactone. Subsequently, 0.5 mg/mL Al(OH)_3_ was added as an adjuvant. Animal immunization, involving rabbits and gerbils, demonstrated significant protection against HTN virus. Moreover, gerbil immunization revealed substantial defense against SEO (Type II) virus attack. The neutralizing antibodies to HTN Hantavirus produced by immunized animals exhibited some cross-reaction to SEO type.

In 2014, Jilin YaTai Biopharmaceuticals Co., Ltd. (China) enhanced the manufacturing process of the hemorrhagic fever vaccine through several optimizations. First, the ultrafiltration purification technology was refined to maximize the removal of cells and other impurities, thereby minimizing adverse reactions post-vaccination. Second, Vero cells were introduced to replace hamster kidney cells, aiming to reduce costs, enhance cell proliferation, and decrease the risk of exogenous contamination. Additionally, improvements were made to the concentration and formulation methods of the bivalent vaccine to ensure product quality and increase manufacturing capacity. Table 2 summarizes the development of inactivated hemorrhagic fever with renal syndrome vaccine.

## 4. Hepatitis A Vaccine

Viral hepatitis A (Hepatitis A) is an acute intestinal infectious disease caused by the infection of the hepatitis A virus, presenting a significant public health concern in developing countries [7]. Two principal hepatitis A strains, TZ84 and Lv 8, have been instrumental in the production of inactivated vaccines. In 1984, the virus was isolated from early stool samples obtained from a patient named Zhang in the rural area of Tangshan, Hebei Province, during an endemic hepatitis A outbreak. Utilizing 2BS cell isolation, culture, and acclimatization passage, the TZ84 strain, suitable for vaccine manufacturing, was carefully selected and cultivated through the 10th generation [8]. In early 1988, the Lv 8 strain was isolated from early stool samples of hepatitis A patients in Lusi Town, Nantong City, by the Institute of Medical Biology, Chinese Academy of Medical Sciences. The strain was subsequently isolated and cultured using KMB17 cells. By the 10th generation, both the infectivity titer (7.0–7.75 logTCID50/mL) and antigen titer (512) met the requisite criteria for vaccine production [9].

Prior to 1996, all inactivated hepatitis A vaccines used in China were imported. To address this domestic gap, the National Institutes for Food and Drug Control (China) collaborated with Beijing Sinovac Biotech Co., Ltd. (China) in a joint research project on the inactivated hepatitis A vaccine. This initiative was included in the “Ninth Five-Year Plan”, a national major scientific and technological research project in the medical and health sector. In 2002, China achieved a significant milestone with the approval for successful marketing of its first inactivated hepatitis A vaccine with independent intellectual property rights, known as Healive^TM^.

Healive^TM^, the hepatitis A inactivated vaccine, was formulated through a systematic process. The HAV TZ84 strain, possessing independent intellectual property rights and utilized as the virus seed, was inoculated onto human embryonic lung diploid cells widely employed in domestic vaccines. The subsequent steps included cultivation, harvesting, purification, formaldehyde inactivation, and aluminum hydroxide adsorption. Noteworthy technological innovations were introduced in the production of Healive^TM^. Initially, both the cell and virus were cultured in an internationally advanced cell factory in China. The cultivated HAV underwent a series of purification steps, including chemical treatment, ultrafiltration, and gel column chromatography, ensuring a high level of antigen purity in the vaccine. The inactivation process involved the use of a 1:4000 formalin solution for 12 days, significantly surpassing the theoretical HAV survival limit. Healive^TM^ Hepatitis A inactivated vaccine meets international standards for similar products, offering unique technological advancements. It is a preservative-free and protective-agent-free safe vaccine. The primary packaging material is a disposable pre-filled syringe with a painless needle, chosen to enhance safety and comfort during inoculation. A robust quality management system was established in adherence to Good Manufacturing Practice (GMP) requirements. Statistical analyses of test results from 16 sub-batches demonstrated a stable manufacturing process with good repeatability and minimal inter-batch differences. The immunogenicity (ED_50_) of HealiveTM hepatitis A inactivated vaccine remained consistent for 33 months at 2–8 °C and 32 days at 37 °C, affirming the vaccine’s stable quality [10].

In 2003, Walvax Biotechnology Co., Ltd. (China) achieved success in identifying a highly efficient and stable proliferating strain of HAV YN5, utilizing Vero cells as the cell substrate. A preliminary study on the trial manufacturing of the inactivated vaccine ensued [11,12]. Subsequently, in 2005, Beijing Sinovac Biotech Co., Ltd. introduced a combined vaccine for Hepatitis A and Hepatitis B. This innovative vaccine comprises an inactivated Hepatitis A vaccine and a genetically engineered Hepatitis B vaccine. Offering simultaneous protection against both infectious diseases, it presents advantages such as reducing the number of vaccinations, alleviating the pain associated with vaccination, and lowering overall vaccination costs. In 2009, Jiangsu Simcere Weike Biopharmaceutical Co., Ltd. obtained the HAV Js-4 strain from the feces of patients with hepatitis A after self-isolation and acclimatization culture. Both animal tests and clinical trials have affirmed that the inactivated hepatitis A vaccine, manufactured using the Js-4 strain as the virus strain and Vero cells as the substrate, exhibits good safety and immunogenicity [13]. Table 3 summarizes the development of hepatitis A vaccine.

## 5. Inactivated Rabies Vaccine for Human Use

Rabies is an acute zoonotic infectious disease affecting the central nervous system and caused by rabies virus infection. Clinical symptoms include hydrophobia, photophobia, dysphagia, mania, and more. Following human exposure, the virus attaches to peripheral nerve terminals and travels to the brain, resulting in an almost 100% fatality rate [14]. Globally, vaccine strains for rabies can be categorized into Pasteur strain (PAS), Flury strain, SAD strain, aG strain, CTN strain, and their derivatives. In China, there are two main rabies vaccine strains with independent intellectual property rights: aG and CTN-1. The aG strain, derived from the “Beijing strain of fixed rabies virus”, originated from a wild strain isolated from rabies-infected dogs’ brains by Yuan Junchang in 1931. It was propagated through rabbit brain, hamster kidney cells, and guinea pig brain [15]. The CTN-1V strain, originating from a rabies patient’s brain tissue in Zibo City, Shandong Province, in 1957, was initially passaged in mouse brain as the CTN-M strain. Subsequently, it underwent passage into CTN-1 strain through human embryo lung diploid cells. Acclimatized to Vero cells, it became the widely used CTN-1V strain in vaccine manufacturing in China [16]. In 1982, China’s Ministry of Health expanded its approval for this virus, and in 1984, the WHO recognized it as a fixed rabies virus approved for vaccine manufacturing. In 1999, Jilin Yatai Biological Pharmaceutical Co., Ltd. (China) obtained approval for the commercialization of a rabies vaccine produced using hamster kidney cells. In 2003, Liaoning Yisheng Biopharma Co., Ltd.’s (China) rabies vaccine produced using Vero cells was approved for market release in China. Many domestic pharmaceutical companies adopted the “CTN-1 strain” and Vero cells for rabies vaccine production. In 2012, Chengdu Kanghua Biological Products Co., Ltd. (China) successfully screened the rabies virus “PM strain” acclimatized to human diploid cells MRC-5, leading to the production of the human diploid cells-derived vaccine (HDCV). The rabies vaccine for human use (PM strain, MRC-5 cells) approved for marketing has elevated the quality of rabies vaccines in China to the international advanced level [17]. Table 4 summarizes the development of inactivated rabies vaccine for human use.

## 6. Inactivated Japanese Encephalitis Vaccine

Japanese encephalitis, a zoonotic ailment, is instigated by the Japanese encephalitis virus, posing a significant threat to the central nervous systems of both humans and animals. It stands as the predominant cause of viral encephalitis in countries across western Asia–Pacific and northern Australia [18]. The P3 strain of the Japanese encephalitis virus was initially isolated from encephalitis samples in China back in 1949. The Beijing Institute subsequently developed an inactivated Japanese encephalitis vaccine in 1950 and 1951 using chicken embryos and mouse brains, respectively. In 1967, the National Vaccine and Serum Institute (Beijing, China) inoculated the P3 strain on hamster kidney cells, leading to the successful development of a hamster kidney cell inactivated vaccine, which was officially manufactured and deployed in 1968 [19].

In 2008, Liaoning Chengda Biotechnology Co., Ltd. (China) achieved a breakthrough by developing a novel generation of inactivated Japanese encephalitis vaccine with independent intellectual property rights in China. This purified vaccine is produced using Vero cells as the cell substrate, ensuring it is mercury-free, liquid, and highly purified. Employing a bioreactor manufacturing process [20]. The vaccine addresses the shortcomings of existing counterparts, offering characteristics such as a high efficiency-cost ratio, good stability, and minimal side effects. The phase 3 clinical study demonstrated that all indicators met international standards, positioning it as an ideal upgraded Japanese encephalitis vaccine in China.

Subsequently, in 2009, Chengdu Kanghua Biological Products Co., Ltd. (China) successfully screened the Japanese encephalitis virus strains P3, SA14-14-2, and Nakayama adapted to human embryonic lung fibroblasts [21,22,23]. Table 5 summarizes the development of inactivated Japanese encephalitis vaccine.

## 7. Inactivated Enterovirus Type 71 Vaccine

Hand, foot, and mouth disease (HFMD) is an infectious ailment caused by various enteroviruses, primarily characterized by lesions on the hands, feet, mouths, and other regions. It tends to affect pre-school children more prominently [24]. In the production of the EV71 inactivated vaccine in China, the main strains utilized belong to the EV71 C4 subtype. This subtype was initially identified during an HFMD outbreak in Taiwan in 1998. Subsequent severe HFMD outbreaks in China’s Hubei and Guangdong regions in 2008 and 2010 were primarily attributed to the C4 subtype. In these outbreaks, the EV71 C4 subtype was isolated from different cell lines to produce M01, H07, and FY strains [25]. In China, three companies have received approval for marketing EV71 inactivated vaccines. The Institute of Medical Biology, Chinese Academy of Medical Sciences, China, cultured the vaccine on diploid KMB17 cells using the M01 strain, and Beijing Sinovac Biotech Co., Ltd. cultured it on Vero cells using the H07 strain. Both were approved for marketing in 2015. The EV71 inactivated vaccine developed by the National Vaccine and Serum Institute in collaboration with the Wuhan Institute of Biological Products Co., Ltd., China, using the FY strain cultured on Vero cells, received marketing approval in 2016. The deployment of this vaccine has effectively curtailed the prevalence and transmission of HFMD in Chinese children. The literature reports indicate a lower incidence of adverse reactions to the EV71 vaccine, with typically mild symptoms and an overall good safety profile [20,26]. Table 6 summarizes the development of inactivated enterovirus type 71 vaccine.

## 8. Inactivated Poliomyelitis Vaccine

Poliovirus, a member of the poliomyelitis pathogen, possesses the capacity to harm motor neurons within the central nervous system, leading to paralysis and potential fatality [27]. The primary strain utilized in China for producing inactivated polio vaccine is the Sabin attenuated strain. The Sabin strain is a weakened form of the wild poliomyelitis virus achieved through multiple passages, acclimatization cultures, and gene recombination. In 1967, the Institute of Medical Biology of the Chinese Academy of Medical Sciences conducted a breeding experiment on four strains of Type III wild polio virus isolated from the feces of healthy children. This effort resulted in the successful selection of an attenuated Type III2 strain in 1969. In 1983, Professor Jiang Shude from the Institute of Biology of the Chinese Academy of Medical Sciences proposed the use of attenuated strains for the production of Inactivated Poliomyelitis Vaccine (IPV). To achieve this, attempts were made to culture the next-generation monkey kidney cells using microcarrier bioreactor culture technology in China. Subsequently, the Institute of Biology of the Chinese Academy of Medical Sciences initiated investigations into virus inactivation and efficacy detection of the Sabin inactivated poliomyelitis vaccine (referred to as “sIPV vaccine”). The utilization of attenuated strains in preparing IPV ensures the safety and effectiveness of the vaccine.

The Sabin strain inactivated polio vaccine represents a significant innovation with entirely independent intellectual property rights in China. In July 2015, the Institute of Medical Biology of the Chinese Academy of Medical Sciences launched the world’s first Sabin inactivated polio vaccine (sIPV), marking the first instance globally and in China of utilizing the current strain (Sabin strain) for manufacturing live attenuated poliomyelitis vaccines. Subsequently, in September 2017, the Beijing Institute officially introduced its version of the sIPV vaccine. This vaccine employed the Sabin poliomyelitis Type I, II, and III strains provided by the World Health Organization, utilizing Vero cells as the cell substrate. In March 2022, the World Health Organization announced that the Sabin strain (Vero cell) inactivated polio vaccine (SIPV) from Sinopharm National Vaccine and Serum Institute in China had successfully undergone pre-certification and was eligible for purchase by the United Nations system. The literature reports affirm the sIPV vaccine’s commendable safety profile, making it suitable for poliomyelitis prevention and vaccination in age-appropriate populations [28]. Table 7 summarizes the development of inactivated poliomyelitis vaccine.

## 9. Quadrivalent Influenza Lysis Vaccine

Influenza, commonly known as the flu, is an acute respiratory infectious disease caused by the influenza virus. The constant genetic variability and host diversity of the influenza virus lead to the emergence of new mutated strains that circulate repeatedly in the population [29]. Early influenza vaccines were produced using chicken eggs. In 2007, Sinovac received approval to market the human H5N1 pandemic influenza vaccine. In 2009, Sinovac achieved a milestone by developing the world’s first influenza A (H1N1) virus vaccine (Split Virion), establishing China’s prominent position in influenza vaccine research. The successful deployment of this vaccine effectively curtailed the spread of pandemic influenza A (H1N1) and yielded positive social benefits. Given the continuous mutation of the virus, zoonotic influenza viruses of the H5N1, H9N2, H7N9, and H10N8 subtypes have been identified as potential pandemic strains in the future. Specific virus strains, designated as vaccine candidates, are identified by the World Health Organization (WHO) based on the global influenza surveillance network. Currently, the trivalent and quadrivalent influenza vaccines are most commonly used. The trivalent vaccine includes two influenza A viruses (H1N1 and H3N2) and one influenza B virus (BV), while the quadrivalent influenza vaccine adds an additional influenza B virus (BY) to the trivalent formulation [30,31]. Since 2018, Hualan Biological Bacterin Inc. (China) and Changchun Changsheng Biotechnology Co., Ltd. China) have successively received approval to market Quadrivalent Influenza Vaccine (Split Virion), Inactivated [32].

In 2022, Sinopharm Wuhan Institute of Biological Products Co., Ltd. (China) successfully completed the phase I clinical study for the quadrivalent influenza virus vaccine (MDCK cells) (Split Virion) with robust support from the Shandong Provincial Center for Disease Control and Prevention. The study demonstrated favorable results, indicating that the vaccine exhibited good safety and immunogenicity. Subsequently, on 28 September 2023, the phase III clinical study commenced in Feicheng, Shandong Province. An innovative aspect of this vaccine is the use of MDCK cells instead of chicken embryos, a departure from the conventional approach approved for quadrivalent influenza virus vaccines (Split Virion) derived from chicken embryos. This adaptation brings forth several advantages, including adherence to strict standards, utilization of new technology, high purity, and a reduced risk of allergic reactions. Notably, this vaccine represents a significant milestone as the first cell substrate influenza vaccine approved for clinical study in China. Table 8 summarizes the development of quadrivalent influenza lysis vaccine.

## 10. Inactivated 2019-nCoV Vaccine

The severe acute respiratory syndrome coronavirus 2 (SARS-CoV-2) is the causative agent of the coronavirus disease 2019 (COVID-19) pandemic. The clinical presentation of the disease varies widely, including asymptomatic cases, individuals with mild or moderate influenza-like symptoms, and those experiencing severe pneumonia, acute respiratory distress syndrome (ARDS), and, in some instances, fatalities attributed to inflammatory factor storms [33]. The original SARS-CoV-2 strain (WIV04), isolated from the bronchoalveolar lavage fluid of a pneumonia patient at Wuhan Jinyintan Hospital, was cultured in Vero E6 cells. Through a virus plaque assay of Vero cells, cloned strains demonstrating high titer, substantial viral particle yield, and genetic stability were selected as inactivated vaccine strains [34,35]. Immunofluorescence detection, virus proliferation detection, evolutionary analysis, and sequence alignment were employed to identify, detect, and analyze the virus strains. These measures ensured that the selected strains met the criteria for vaccine strains. Other companies, including Sinovac Biotech and Beijing Biolegend (China), undertook the cloning and screening of isolated clinical virus strains. Vaccine candidate strains were determined based on comprehensive considerations, taking into account factors such as virus proliferation, immunogenicity, antigen expression, and potential adventitious agents [35,36,37,38,39,40].

In April 2020, the Wuhan Institute of Biological Products achieved the initial development of the 2019-nCoV vaccine, inactivated (Vero cells). Subsequently, on 30 December 2020, the 2019-nCoV vaccine, manufactured by the National Vaccine and Serum Institute, received conditional marketing approval in China and obtained emergency use authorization from the WHO on 7 May 2021. In February 2021, the CoronaVac inactivated vaccine for the new coronavirus, developed by Beijing Kexing Zhongwei Biological Technology Co., Ltd., China, was conditionally listed in China and added to the WHO’s emergency use list in June 2021. As of now, four technical approaches to vaccines are available to address the COVID-19 outbreak: inactivated vaccine, mRNA vaccine, viral vector vaccine, and protein vaccine [41]. Each vaccine type has its own set of advantages and disadvantages. Viral vector and protein vaccines, when compared to mRNA and inactivated vaccines, fall in between in terms of protective efficacy, safety, and accessibility. In China, inactivated vaccines are predominantly used, while mRNA vaccines are more prevalent in developed countries in Europe and the United States.

According to literature reports, from 14 December 2020 to 14 June 2021, during the study period, the United States was vaccinated with 298,792,852 doses of mRNA vaccine. VAERS handled 340,522 reports: 313,499 (92.1%) were non-serious reports, 22,527 (6.6%) were serious reports (non-fatal) and 4496 (1.3%) were fatal reports [42]. China’s inactivated COVID-19 vaccine has demonstrated notable safety based on monitoring adverse reactions. The reported rate of adverse reactions was slightly lower than that of routine annual vaccinations. By 30 May 2022, over 3.38 billion doses of the inactivated 2019-nCoV vaccine had been administered nationwide, resulting in 238,215 reported adverse events, equating to an overall incidence of 70.45 cases per 1 million doses. Notably, countries employing China’s inactivated vaccines extensively have not identified any safety issues. These findings substantiate the high safety profile of China’s inactivated 2019-nCoV vaccines.

mRNA vaccines necessitate extremely low storage temperatures (−40 °C to −70 °C), posing logistical challenges in cold chain management. Developing countries face constraints in purchasing power and vaccine storage conditions. Inactivated vaccines offer notable advantages in cold chain logistics as they can be stored and transported at 2 °C to 8 °C, aligning with the existing storage capabilities of many countries. This obviates the need for substantial cold chain infrastructure modifications. Although inactivated vaccines demonstrate lower efficacy in preventing mild COVID-19 cases compared to mRNA vaccines, their superior safety and accessibility make them well-suited for widespread adoption in developing countries. Table 9 summarizes the development of the inactivated 2019-nCoV vaccine.

## 11. Challenges and Prospects

### 11.1. Challenges

In the realm of inactivated vaccines, China has achieved significant breakthroughs, contributing to the assurance of public health and readiness against emerging health threats. Inactivated virus vaccines, a cornerstone in preventing various viral diseases such as influenza, hepatitis, and poliomyelitis, have been successfully developed and manufactured.

Nevertheless, the development of inactivated virus vaccines encounters challenges stemming from the rapid evolution of viruses and the emergence of new vaccines. First, antibody-dependent disease enhancement (ADE) heightens the virus’s ability to enter host cells, potentially exacerbating diseases [43]. Known ADE effects have been observed in human viruses such as dengue, influenza, Ebola, SARS-CoV, and MERS-CoV [44,45,46,47,48,49,50]. This implies that vaccinated individuals may have antibodies that bind to the virus but do not completely eliminate it, increasing the virus’s ability to enter host cells and worsening the disease. Second, the swift mutation capacity of RNA viruses, exemplified by influenza and 2019-nCoV, presents a challenge in developing relevant vaccines. To counteract this mutation, continuous vaccine updates are required in order to ensure effectiveness against the latest virus variants. For instance, after the 2019 COVID-19 outbreak, SARS-CoV-2 produced WHO-defined variants of concern (VOCs) such as Alpha, Beta, Gamma, Delta, and Omicron [51,52]. Third, viruses exhibit widespread complexity and immune evasion abilities. Examples include influenza virus mutating through antigenic changes, HIV hiding in host cells to evade the immune system, and the herpes virus escaping immune clearance through latent infection [53,54,55]. Lastly, the rapid advancement of new vaccine technologies, such as genetically engineered subunit vaccines, genetically engineered live vector vaccines, and mRNA vaccines, poses challenges to traditional inactivated vaccines [56,57,58,59,60].

### 11.2. Prospects

Despite the obstacles faced, inactivated virus vaccines remain highly promising. Newer vaccines may lack immunogenicity or carry safety risks, whereas the long-standing history and mature technology of inactivated virus vaccines have thoroughly validated their safety and efficacy. This positions them as a crucial tool in preventing infectious diseases. The prospective application of inactivated virus vaccines is evident in the following.

#### 11.2.1. Development of Novel Cell Substrates

To better align with specific manufacturing processes, manufacturers are innovating by developing novel cell substrates [61]. Noteworthy patented cell lines, including PER.C6^®^, AGE1.CR.pIX, and EB66^®^, as well as PBS-12SF, DuckCelt^®^-T17, and MFF-8C1, have been reported. A majority of these cell lines are amenable to serum-free suspension culture.

PER.C6^®^, originated by Crucel in Netherlands, is derived from human embryonic retinal cells and exhibits sensitivity to adenovirus, influenza virus, poliovirus, and Ebola virus. In Belgium, the initial clinical study assessed the Sabin inactivated poliovirus vaccine (sIPV) based on a high-yield PER.C6^®^ cell line. Results demonstrated good tolerability and high immunogenicity in adults with anti-poliovirus antibodies [62]. Subsequent evaluations in infants revealed that sIPV based on PER.C6^®^ cells induced high serum conversion rates and geometric mean titers of neutralizing antibodies against all three Sabin strains, maintaining acceptable safety and immunogenicity [63].

AGE1.CR.pIX, developed by ProBioGen in Germany, is derived from duck retina cells and demonstrates sensitivity to smallpox virus, fowlpox virus, and influenza virus. Currently, Vaccitech in Britain has utilized AGE1.CR.pIX to develop the universal influenza A vaccine MVA-NP + M1 [64]. Research by Trabelsi et al. [65] indicates that in the AGE1.CR.pIX suspension cell line, the use of chemically defined media can effectively produce a scalable process for rabies vaccine intended for animal use.

EB66^®^, developed by Valneva in France, originates from duck embryo cells and exhibits sensitivity to influenza, measles, mumps, and poxviruses. In 2014, an EB66^®^ cell-based H5N1 pandemic influenza reserve vaccine received marketing approval in Japan [66]. The H5N1 influenza vaccine (AS03 adjuvant) manufactured using the EB66^®^ cell culture platform (KD-295), as developed by Endo et al. [67], demonstrated good tolerability and high immunogenicity. Nikolay et al. [68] reported that EB66^®^ perfusion culture effectively increased the yield of flaviviruses and Zika virus.

Coussens et al. [69] introduced an immortalized chicken embryo cell line named PS-12SF, well-suited for serum-free growth and proficient in replicating both human and recombinant H5N1 influenza strains to high titers. In numerous instances, the influenza virus growth titer in PBS-12SF cells surpassed that in primary chicken embryo kidney cells, MDCK cells, and Vero cells. Notably, in PBS-12SF cell culture, influenza virus release into the culture media occurred without the need for exogenous proteases, simplifying downstream processes in vaccine manufacturing.

The DuckCelt^®^-T17 cell line, derived from primary duck embryonic cells expressing duck telomerase reverse transcriptase, exhibited a maximum density of 6.5 × 10^6^ mL^−1^ in batch suspension culture under serum-free conditions. The culture volume could be scaled up from 10 mL to a 3 L bioreactor. Through optimization of infection conditions, DuckCelt^®^-T17 cell lines facilitated the production of multiple human, avian, and swine influenza viruses with high infection titers (>5.8 lgTCID_50_/mL) [70].

Dong et al. [71] introduced a novel fibroblast-like cell line called MFF-8C1, derived from the early primary culture of Mandarin fish fry through single-cell cloning. MFF-8C1 cells exhibited robust growth in Dulbecco Modified Eagle medium containing 10% fetal bovine serum. Moreover, the cultured CMV suspension demonstrated high toxicity to infected Mandarin fish, indicating MFF-8C1 as a promising candidate cell substrate for CMV vaccine manufacturing.

#### 11.2.2. Development of Novel Virus Inactivation Modalities and Novel Adjuvants

Virus inactivation plays a pivotal role in ensuring the safety of the final product. Conventional inactivated vaccines typically employ chemical inactivation methods, such as formaldehyde and β-propanolide, to alter the structure of the virus’s protein or genetic material, rendering it incapable of causing infection. However, these traditional chemical inactivation approaches elevate the risk of residual chemical reagents in the product, thereby increasing safety concerns and manufacturing costs. Consequently, the development of novel virus inactivation methods becomes imperative to enhance product quality and mitigate manufacturing risks. A recent study by RAGAN et al. [72] introduced a new inactivation approach for crafting the 2019-nCoV vaccine, inactivated. This method utilizes riboflavin in conjunction with ultraviolet irradiation to modify the structure of viral nucleic acid. The resulting SARS-CoV-2 totivirus inactivated vaccine exhibited a protective effect in hamster challenge models. Notably, the use of riboflavin in this process demonstrated good safety characteristics, free from mutagenicity and carcinogenicity, and avoided issues such as alkylation, crosslinking, and covalent modifications [73]. In a related investigation, Karakus et al. [74] employed γ-inactivated SARS-CoV-2 vaccine (OZG-3861–01) to stimulate the production of neutralizing antibodies in BALB/c mice. Importantly, no antibody-dependent enhancement of disease (ADE) was observed in the study.

A vaccine adjuvant serves to nonspecifically modify or enhance the specific immune response to antigens, playing a supportive role. Adjuvants can stimulate the production of a sustained and effective specific immune response, improving the body’s protective capabilities. This, in turn, allows for a reduction in the quantity of immune substances required, leading to a decrease in vaccine manufacturing costs. Xu Chuanlai’s research team at Jiangnan University uncovered that a distinctive chiral nano immune adjuvant could equally elicit both humoral and cellular immune responses. The team systematically screened various chiral ligands, incorporating polarized light during the nano adjuvant preparation process. By optimizing the combination with different wavelengths of polarized light to induce symmetry breaking on the high-index crystal surface, they achieved a uniform surface topography. With an anisotropy factor reaching 0.44, the team successfully synthesized highly chiral nanoadjuvants, offering not only theoretical backing for protective vaccine development but also indicating a pathway for therapeutic vaccine advancement [75]. Additionally, a novel aluminum-containing composite adjuvant known as AS04, developed internationally, has received approval for use in the HBV vaccine Fendrix and the HPV vaccine Cervarix [76,77].

#### 11.2.3. Development of Polyvalent Inactivated Vaccines

Polyvalent vaccines offer more effective and broad-spectrum protection against various pathogens, exemplified by the quadrivalent influenza vaccine and the nine-valent HPV vaccine. Despite this, there is currently no approved polyvalent 2019-nCoV vaccine for marketing globally or domestically. To enhance the body’s efficient immunity against multiple mutant strains, decrease the number of vaccinations, and effectively address the ongoing Delta and Omicron mutant strains pandemic, the development of a polyvalent 2019-nCoV vaccine, such as trivalent (wild type, Delta, and Omicron), holds significant importance. While monovalent vaccines have demonstrated protective effects against specific mutant strains, they are limited to individual strains and cannot effectively combat the current circulating mutant strains and potential future variants. Polyvalent inactivated vaccines, on the other hand, can mitigate the physical and psychological discomfort associated with multiple injections, enhance vaccination compliance, simplify vaccine management, reduce vaccination and management costs, and minimize vaccine-related adverse reactions. These factors collectively contribute to an improved vaccination rate, indirectly reducing the overall public health costs for society. Thus, the development of a polyvalent inactivated 2019-nCoV vaccine holds great promise for enhancing effective immunity against multiple variant strains while streamlining vaccination procedures [78].

#### 11.2.4. A Variety of Means to Reduce Manufacturing Costs

In addition to ensuring safety and effectiveness, the consideration of manufacturing cost is crucial for the widespread adoption of vaccines. Several approaches can be employed to lower the manufacturing cost of inactivated virus vaccines. First, enhancing manufacturing efficiency is pivotal. This can be achieved through the utilization of automated equipment, modern production lines, and the application of AI. These technologies demonstrate promising prospects for improving the overall efficiency of vaccine production. Second, optimizing the manufacturing process is essential. By refining the inactivation process through the implementation of more effective methods to eliminate the virus and minimizing raw material wastage, manufacturing costs can be significantly reduced. Lastly, reducing the cost of raw materials is a key factor. Since inactivated virus vaccines require a substantial amount of viruses as raw materials, cost reduction can be achieved by developing more efficient virus screening methods and employing more economical media for production. This approach contributes to an effective reduction in overall manufacturing expenses. Manufacturing scale-up: the expansion of manufacturing operations has the potential to lower vaccine production costs; for instance, utilizing scale-up techniques can lead to cost advantages, ultimately decreasing manufacturing expenses. Policy Assistance: governments can improve vaccine production through measures such as subsidies, tax relief, and other support mechanisms.

Initially, advancements in creating innovative cell substrates have contributed to the ongoing enhancement of cell culture technology. This progress has resulted in increased manufacturing efficiency and cost-effectiveness, thereby fostering the continual improvement of vaccines. Subsequently, refining the inactivation process and optimizing adjuvants has further enhanced the immunogenicity of inactivated vaccines. Additionally, the creation of polyvalent inactivated vaccines has positively impacted vaccination acceptance. Lastly, efforts to enhance the accessibility of inactivated vaccines involve vaccine manufacturers implementing diverse strategies to reduce the manufacturing costs associated with these vaccines.

## Figures and Tables

**Table 1 pharmaceutics-15-02721-t001:** Development of inactivated tick-borne encephalitis vaccine.

Time (Year)	Production Unit	Cell Line	Stage of Development
1958	Changchun Institute of Biological Products Co., Ltd.	Chick embryo cell	Chicken embryo cells replace mouse brain tissue
1967	Changchun Institute of Biological Products Co., Ltd.	Hamster kidney cell	Hamster kidney cells replace chicken embryo cells
2001	Changchun Institute of Biological Products Co., Ltd.	Hamster kidney cell	Continuous flow centrifugation, ultrafiltration concentration, and column chromatography purification were used to develop a purified vaccine

**Table 2 pharmaceutics-15-02721-t002:** Development of inactivated hemorrhagic fever with renal syndrome vaccine.

Time (Year)	Production Unit	Cell Line	Stage of Development
2002	Changchun Institute of Biological Products Co., Ltd.	Hamster kidney cell	Bivalent vaccine replace Monovalent vaccine
2003	Royal (Wuxi) Biopharmaceuticals Co., Ltd.	Vero cell	Vero cell replace Hamster kidney cell
2005	Zhejiang Tianyuan Biopharmaceuticals Co., Ltd.	Gerbil kidney cell	Gerbil kidney cell replace Hamster kidney cell
2014	Jilin YaTai Biopharmaceuticals Co., Ltd.	Vero cell	Improve the ultrafiltration purification technology and improve the concentration and deployment method of bivalent vaccine

**Table 3 pharmaceutics-15-02721-t003:** Development of hepatitis A vaccine.

Time (Year)	Production Unit	Cell Line	Stage of Development
2002	National Institutes for Food and Drug Control and the Beijing Sinovac Biotech Co., Ltd.	2BS cell	The first inactivated hepatitis A vaccine with independent intellectual property rights was marketed
2003	Walvax Biotechnology Co., Ltd.	Vero cell	Vero cells replace human embryonic lung diploid cells and are still under investigation
2005	Beijing Sinovac Biotech Co., Ltd.	2BS cell	Hepatitis A and B combined vaccine

**Table 4 pharmaceutics-15-02721-t004:** Development of inactivated rabies vaccine for human use.

Time (Year)	Production Unit	Cell Line	Stage of Development
1999	Jilin Yatai Biological Pharmaceutical Co., Ltd.	Hamster kidney cell	First domestic listing
2003	Liaoning Yisheng Biopharma Co., Ltd.	Vero cell	Vero cells replace hamster kidney cells
2012	Chengdu Kanghua Biological Products Co., Ltd.	Human diploid cell	Human diploid cells replace Vero cells

**Table 5 pharmaceutics-15-02721-t005:** Development of inactivated Japanese encephalitis vaccine.

Time (Year)	Production Unit	Cell Line	Stage of Development
1968	National Vaccine and Serum Institute	Hamster kidney cell	First domestic listing
2008	Liaoning Chengda Biotechnology Co., Ltd.	Vero cell	Vero cells replace hamster kidney cells
2009	Chengdu Kanghua Biological Products Co., Ltd.	Human diploid cell	Human diploid cells replace hamster kidney cells and are still under investigation

**Table 6 pharmaceutics-15-02721-t006:** Development of inactivated enterovirus type 71 vaccine.

Time (Year)	Production Unit	Cell Line	Stage of Development
2015	Institute of Medical Biology, Chinese Academy of Medical Sciences	KMB17 cell	The world’s first inactivated EV71 vaccine
2015	Beijing Sinovac Biotech Co., Ltd.	Vero cell	Vero cells replace KMB17 cells

**Table 7 pharmaceutics-15-02721-t007:** Development of inactivated poliomyelitis vaccine.

Time (Year)	Production Unit	Cell Line	Stage of Development
2015	Institute of Medical Biology, Chinese Academy of Medical Sciences	Vero cell	The world’s first sIPV vaccine
2017	Sinopharm National Vaccine and Serum Institute	Vero cell	Pre certification by WHO in 2022

**Table 8 pharmaceutics-15-02721-t008:** Development of quadrivalent influenza lysis vaccine.

Time (Year)	Production Unit	Cell Line	Stage of Development
2007	Beijing Sinovac Biotech Co., Ltd.	Chick embryo	China’s first human H5N1 pandemic influenza vaccine
2009	Beijing Sinovac Biotech Co., Ltd.	Chick embryo	The world’s first vaccine for influenza A (H1N1) virus cleavage
2018	Hualan Biological Bacterin Inc.	chick embryo	Qurivalent influenza lysis vaccines replace monovalent vaccines
2022	Wuhan Institute of Biological Products	MDCK cell	MDCK cells replaces chick embryo

**Table 9 pharmaceutics-15-02721-t009:** Development of inactivated 2019-nCoV vaccine.

Time (Year)	Production Unit	Cell Line	Stage of Development
April 2020	Wuhan Institute of Biological Products	Vero cell	The world’s first inactivated 2019-nCoV Vaccine
December 2020	National Vaccine and Serum Institute	Vero cell	Obtained WHO emergency use authorization on 7 May 2021
March 2021	Beijing Kexing Zhongwei Biological Technology Co., Ltd. (China)	Vero cell	Obtained WHO emergency use authorization in June 2021

## Data Availability

Not applicable.

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
