# Peer review of "30-Year Development of Inactivated Virus Vaccine in China"

_pharmaceutics, 2023, doi:10.3390/pharmaceutics15122721_

Round 1
Reviewer 1 Report
Comments and Suggestions for Authors
The manuscript of Shi et al. aims to describe the development of inactivated viral vaccines in China over the last three decades. This is a very hectic and poorly organized review, which is extremely difficult to read and hardly useful in the present form. It also suffers from poor English and wide-spread use of non-scientific language seen from the very beginning ('nurtured' instead of 'passaged' or 'grown' in line 34 and 'sieving' instead of 'selecting' in line 35). It may actually get worse to the point of no comprehension like in line 51: 'virus was infected with a dense monolayer of Vero cells.' Authors just do not know what they are writing about or are completely careless with words since it is the cell line, which is infected with the virus, not the other way around. The overall structure of the review: superficial viral strain description followed by cell lines (not 'substrates' as in text!), and then by 'Production Process' and absolutely empty 'Innovations' does not make sense since many of these sections are filled with material, which has no relevance to the stated title (Production of inactivated vaccines in China), including e.g., section on Covid-19 (inactivated vaccine against Covid-19 is poorly protective and pretty much useless). Actually, the description of various strains in the first section could be the most interesting, but it has no rationales for strain selection and limited information on how this or that strain is currently used for vaccine production (it seems that many of them are not). Authors should completely rewrite the manuscript focusing only on those inactivated vaccines that are currently produced and licensed in China and to describe how different or similar they are from vaccines from other entities. This should be done on disease-by-disease fashion, not in very liberal piecemeal fashion. Professional writing help should be requested to ensure the quality of English, including appropriate use of scientific terms. Not ready for the publication in the present form.
Comments on the Quality of English LanguagePoorly written, many non-scientific terms, syntax mistakes.
Author Response
1.Authors should completely rewrite the manuscript focusing only on those inactivated vaccines that are currently produced and licensed in China and to describe how different or similar they are from vaccines from other entities.
Answer:The manuscript has been rewritten with modifications to the overall structure of the review,and an introduction provided on the main inactivated vaccine products and their development process.
- 'Production Process' and absolutely empty 'Innovations' does not make sense since many of these sections are filled with material, which has no relevance to the stated title (Production of inactivated vaccines in China).
Answer:The "production process" and vague "innovation" have been removed,while emphasis on the prospects has been strengthened.
3.This should be done on disease-by-disease fashion, not in very liberal piecemeal fashion.
Answer:The article introduces each vaccine in chronological order, providing a timeline of their development and implementation.
4.inactivated vaccine against Covid-19 is poorly protective and pretty much useless.
Answer:However, I do not agree with your statement that "inactivated vaccine against Covid-19 is poorly protective and pretty much useless." According to the analysis of Phase III clinical trial data of Sinopharm's inactivated COVID-19 vaccine, it has shown good safety after vaccination by China National Biotec Group's Beijing subsidiary. After two doses of the immunization program, the vaccine recipients in the trial produced high-titer antibodies, with a seroconversion rate of neutralizing antibodies being 99.52%. The vaccine demonstrated an efficacy of 79.34% in preventing COVID-19, caused by infection with the novel coronavirus, which meets the technical standards set by the World Health Organization's "COVID-19 Vaccines: Technical Specifications and Guidelines for Emergency Use Listing" and the related standards required by China's National Medical Products Administration in the "Guidance Principles for Clinical Evaluation of COVID-19 Preventive Vaccines (Trial)".
Reviewer 2 Report
Comments and Suggestions for Authors
This is a very good review describing the state of the art in inactivated vaccines in China, which is of great interest for the field worldwide. The manuscript is well organized and clear when read. I have only one suggestion: to add some information about the necessity (or not) to boost the anti-covid-19 inactivated vaccines, in comparison to mRNA vaccines, as in most countries where these latter have been used, several doses have been administered to the population.
Author Response
Reviewer: 2
Major Comments:
1.I have only one suggestion: to add some information about the necessity (or not) to boost the anti-covid-19 inactivated vaccines, in comparison to mRNA vaccines, as in most countries where these latter have been used, several doses have been administered to the population.
Answer:Thank you for your appreciation. I will answer the necessity of promoting and using inactivated COVID-19 vaccines in developing countries from three perspectives. Firstly, in terms of efficacy, Sinopharm's inactivated COVID-19 vaccine has shown a 79.34% efficacy in preventing COVID-19, meeting the requirements of the World Health Organization's technical standards for emergency use listing of vaccines. Prior to this, mRNA vaccines developed by Pfizer and Moderna in the United States have also reported efficacy rates exceeding 90%. It may seem that our vaccine has a lower efficacy, but this is not because our technology is inferior. It is mainly due to the inherent differences between mRNA and inactivated vaccines. Inactivated virus vaccines have relatively weaker immunogenicity and lower effectiveness compared to mRNA vaccines, which is not surprising. Secondly, in terms of safety, on March 1, 2022, the U.S. Food and Drug Administration was forced to release the review documents for the Pfizer vaccine. Key information in these documents, such as a "death rate of up to 2.9%," "nine pages of side effects," "over 1,000 types of side effects," and "42,000 deaths out of 1,223 cases," has raised doubts among global netizens about the Pfizer mRNA vaccine. On the other hand, the inactivated COVID-19 vaccine in our country has demonstrated extremely high safety in terms of monitoring adverse reactions. The reported rate of adverse events after vaccination is even slightly lower than that of routine annual vaccination. As of May 30, 2022, over 3.38 billion doses of the inactivated COVID-19 vaccine have been administered nationwide, with a total of 238,215 reported adverse events, resulting in an overall reporting rate of 70.45 cases per million doses. Furthermore, countries that have extensively used our country's inactivated COVID-19 vaccine have not reported any vaccine safety issues. These data fully demonstrate the high safety of our inactivated COVID-19 vaccine. Lastly, in terms of cost, the official price for a dose of the inactivated COVID-19 vaccine in China is 200 CNY (27.3360 USD), while the price of mRNA vaccines varies in different countries and regions, generally ranging from 20 USD to 40 USD. The difference between the two is not significant. However, mRNA vaccines require extremely low storage and transportation temperatures (-40 to -70℃), posing high requirements for cold chain logistics. Developing countries have limited purchasing power and vaccine storage and transportation conditions compared to developed countries. Inactivated vaccines have obvious advantages in cold chain transportation, with storage and transportation conditions of 2℃ to 8℃, consistent with the existing vaccine storage and transportation levels in many countries, thus not requiring significant reconstruction of cold chain facilities, which makes them more accessible. In conclusion, compared to mRNA vaccines, inactivated COVID-19 vaccines have better safety and accessibility, making them suitable for promotion and use in developing countries.
Reviewer 3 Report
Comments and Suggestions for Authors
The authors set out to provide a comprehensive, accurate and useful review of the use of the development and use of inactivated virus vaccines in China over the past 30 years. I believe that they have achieved their stated aim.
I have found this review to be comprehensive and accurate, and it provides a very useful history of vaccine production in China over three decades and in several cases, much earlier developments are mentioned.
The review is arranged into a logical order in which viruses causing particular diseases in humans are treated, commencing with Tick-borne Encephalitis (TBE) and progressing through to the most recently encountered Severe Acute Respiratory Syndrome Coronavirus 2 (SARS-CoV-2) responsible for the COVID-19 pandemic.
Some of the nine diseases treated such as TBE (isolated in 1952, vaccine produced 1953) and Human Rabies Virus (isolated 1931, vaccine produced 2005) were isolated long ago however development of approved safe and effective vaccines took many years to appear. All of these diseases precipitated the development of safe and effective vaccines in China but the appearance of the COVID-19 pandemic in 2019 stimulated massive international efforts both in China and in many laboratories worldwide.
The review describes the great innovations and discoveries made in laboratories in China both before the appearance of COVID-19 and before this time.
A detailed account is provided of the role of cell substrates in the production if inactivated vaccines. This section uses sub headings to enable the reader to progress through Primary Cells (Chicken embryo and Gerbil kidney cells), Passaged Cells (Vero cells and MDCK cells) then Human Diploid Cells. Highlights appear at intervals in the review such as mention of the clinically significant report of Fan Bin et al (2019) that adverse reaction rates from administering rabies vaccines produced using human diploid cells were lower than those using Vero cell-purified rabies vaccines.
The authors, quite fairly, describe both the benefits and limitations in the use of diploid cell substrates in large-scale industrial culture vaccine production. The benefits are clear, but significant challenges remain and, in the future, it will be interesting to see further developments toward use of bioreactors to achieve the large scale capacity for vaccine production as we surely will face future challenging pandemics.
The review presents in useful detail, progress in China with improvements in cell culture and virus culture technologies. Advances in virus inactivation techniques are briefly covered, as are advances in antigen separation and purification.
Section 4 covers advances within China relating to developments with inactivated vaccines. Recent developments with particular inactivated vaccines include Inactivated TBE Vaccines, Inactivated Hepatitis A Vaccines, Inactivated Japanese Encephalitis Vaccines, Inactivated Enterovirus 71 (EV71) Vaccine, Inactivated Poliovirus Vaccines and Inactivated COVID-19 Vaccine. For each of the listed diseases, Chinese laboratories have recently produced widely accepted and effective vaccines.
The concluding Section 5, Challenges and Prospects, briefly discusses some of the problems associated with the development of inactivated virus vaccines. In many cases these work with great effect however in diseases where the virus may mutate additional problems arise. Some of these problems are discussed.
The authors also conclude by mentioning multivalent vaccines that have an established place in vaccination programs, providing protection against an array of pathogens. Multivalent vaccines could certainly help address and reduce the incidence of vaccine hesitancy.
Minor corrections, questions and suggestions
I find that with reviews a well-designed flow diagram can, in one page, convey the links between important headings. With the review period spanning three decades there are some interesting stages reached before a final effective vaccine is produced. There are many ways to do this but the content is there in this review to construct such a flow diagram. Clearly this is not essential but I believe it would add to the message contained in the review!
L136 I suggest using an alternative word to triumphant – alternative maybe widely acknowledged
L261 0.30x 106 to 0.50 x 106 cells/cm2 .. Should the 106 appear as 10exp6?
L265 …and 7.50% one month after booster… Should this be 75.00%??
L274 Suggesting a change to sentence: “For safety reasons, due to the tumorigenic nature of tumor tissues, passaged cells employed for vaccine production typically originate from normal tissues.” eg Avoids having to use possessive - tissue’s.
L333 The observation of Fan Bin et al (2019) regarding benefits in achieving lower adverse reaction rates in administering rabies vaccines produced using human diploid cells compared with those following Vero cell-purified rabies vaccines is notably of clinical significance when there is internationally, concern with achieving low adverse reaction rates.
Comments on the Quality of English LanguageThe quality of the English Language used in this review is very good.
Author Response
Major Comments:
1.I find that with reviews a well-designed flow diagram can, in one page, convey the links between important headings. With the review period spanning three decades there are some interesting stages reached before a final effective vaccine is produced. There are many ways to do this but the content is there in this review to construct such a flow diagram. Clearly this is not essential but I believe it would add to the message contained in the review!
Answer:Due to strong demands for rewriting the manuscript from other editors, extensive changes have been made to the structure and sentences of the article. Adding a diagram is a good suggestion, and we took it.Thank you for your suggestion!
Round 2
Reviewer 1 Report
Comments and Suggestions for Authors
The manuscript became much better and can be published after minor editing and spell-check. E.g., something is wrong with the title of Section 11: "11. Challenges and." And what? Please, be more attentive the next time around.
Comments on the Quality of English LanguageJust a minor read-through to verify that all the terms/tenses are correct.
Author Response
We are very grateful to the reviewers for careful and thorough reading of this manuscript and for the thoughtful comments and constructive suggestions, which help to improve the quality of this manuscript. Our response follows:
Major Comments:
The manuscript became much better and can be published after minor editing and spell-check. E.g., something is wrong with the title of Section 11: "11. Challenges and." And what? Please, be more attentive the next time around.
Reply: We have corrected the incorrect title 11, the revised title is "11. Challenges and Prospects" and "11.2 Prospects", furthermore, we checked the spelling, punctuation and expression of the whole text to ensure the quality of English. Thank you again for your suggestion.
